# Detection of Human Herpesviruses in Sera and Saliva of Asymptomatic HIV-Infected Individuals Using Multiplex RT-PCR DNA Microarray

**DOI:** 10.3390/pathogens12080993

**Published:** 2023-07-28

**Authors:** Irna Sufiawati, Rahmi Harmiyati, Nanan Nur’aeny, Agnes Rengga Indrati, Ronny Lesmana, Rudi Wisaksana, Riezki Amalia

**Affiliations:** 1Department of Oral Medicine, Faculty of Dentistry, University of Padjadjaran, Bandung 40132, Indonesia; nanan.nuraeny@fkg.unpad.ac.id; 2Oral Medicine Residency Program, Faculty of Dentistry, University of Padjadjaran, Bandung 40132, Indonesia; rahmiharmiyati05@gmail.com; 3Department of Clinical Pathology, Faculty of Medicine, University of Padjadjaran, Bandung 45363, Indonesia; agnesariantana.sppk@gmail.com; 4Department of Biomedical Sciences, Faculty of Medicine, University of Padjadjaran, Bandung 45363, Indonesia; ronny@unpad.ac.id; 5Department of Internal Medicine, Faculty of Medicine, University of Padjadjaran, Dr. Hasan Sadikin Central General Hospital, Bandung 45363, Indonesia; rudiw98@gmail.com; 6Department of Pharmacology and Clinical Pharmacy, Faculty of Pharmacy, University of Padjadjaran, Bandung 45363, Indonesia; riezki.amalia@unpad.ac.id

**Keywords:** HHVs, HIV, saliva, serum, multiplex PCR

## Abstract

Human herpesviruses (HHVs) are frequently linked to an increased risk of acquiring human immunodeficiency virus (HIV), and vice versa. This study aimed to detect human herpesvirus (HHV) members in the sera and saliva of asymptomatic HIV-infected individuals. Paired saliva and serum samples were obtained from 30 asymptomatic HIV-infected individuals. HHVs were detected with a multiplex reverse transcription-polymerase chain reaction (RT-PCR) DNA microarray Clart^®^Entherpex kit. A total of 30 subjects were enrolled: 23 (76.67%) men and 7 (23.33%) women. The present study showed that at least one or more HHV members were detected in the saliva and sera of all (100%) of the subjects. In the saliva, we detected herpes simplex virus 1 (HSV-1) 6.67%, herpes simplex virus 2 (HSV-2) 6.67%, Epstein–Barr virus (EBV) 86.67%, cytomegalovirus (CMV) 63.33%, HHV-6 (40%), and HHV-7 (83.33%). In the sera, HSV-2 (20%), EBV (30%), CMV (40%), HHV-6 (0%), and HHV-7 (76.67%) were found, but not HSV-1. VZV and HHV-8 were not detected in either the saliva or sera. EBV and HHV6 were significantly more prevalent in the saliva than they were in the sera of asymptomatic HIV-infected individuals (*p* < 0.05). However, no significant differences were found in the prevalence of HSV-1, EBV, CMV, HHV-6, and HHV-7 in the saliva and sera of asymptomatic HIV-infected individuals (*p* > 0.05). In conclusion, the multiplex RT-PCR DNA microarray can serve as a valuable diagnostic tool that can be used as a screening tool or a first-line test for HHVs infections.

## 1. Introduction

The human herpesviruses (HHVs) consist of herpes simplex virus 1 (HSV-1), herpes simplex virus 2 (HSV-2), varicella zoster virus (VZV), Epstein–Barr virus (EBV), cytomegalovirus (CMV), HHV-6, and HHV-7, and Kaposi’s sarcoma-associated herpesviruses (KSHV or HHV-8) are the most prevalent opportunistic viral infections in human immunodeficiency virus (HIV)-infected individuals [1,2]. The critical role of HHVs as co-pathogens in the progression of HIV has been observed [3]. These viruses can also cause various diseases in the oral cavity that may frequently recur, be persistent, induce severe symptoms, and be life threatening for HIV-infected individuals [4]. In the combined antiretroviral therapy (cART) era, HHV-associated oral diseases in HIV-infected individuals are significantly prevalent and remain a challenge [5]. Antiviral medications are effective against active HHV infections. However, it is important to note that no standardized antiviral treatment has been specifically approved for HHV-6 or EBV infections [6].

In the oral cavity of HIV-infected individuals, reactivation may provide a favorable microenvironment for the interaction between HIV and HHVs. The interaction between HHV members and HIV within the oral mucosal epithelium may lead to the disruption of the oral epithelial barrier and promote the paracellular penetration and spread of HHVs, which may result in abundant infectious virions shedding into the saliva [7,8,9,10,11]. Several researchers have investigated members of HHVs from the saliva of HIV-infected individuals [12,13,14,15].

Screening for HHVs in asymptomatic individuals living with HIV can be instrumental in preventing disease progression. The detection of HHVs is also important for clinical diagnosis and treatment because HHVs infections may not be accurately diagnosed during a clinical examination; therefore, laboratory diagnosis should be conducted. Multiplex PCR can serve as a valuable diagnostic tool for HHVs infections to rapidly and simultaneously detect all HHV members or other pathogens in a clinical specimen, and this can reduce the cost and time. It is also equally as sensible and specific as reference tests are [16,17].

Previously, low prevalence rates of herpesvirus DNA were detected in saliva samples from a healthy population using a multiplex polymerase chain reaction (PCR) [18]. However, there are still limited amounts of data available on the simultaneous presence of HHVs in the salivary samples of HIV-infected individuals. The aim of this study is to determine the frequency of HHVs in saliva from HIV-infected individuals and to compare the patterns in their blood using an mRT-PCR DNA microarray.

## 2. Materials and Methods

This cross-sectional study was conducted in an outpatient HIV clinic in Hasan Sadikin General Hospital, the main referral hospital in West Java, Indonesia. Purposive sampling was conducted using the following inclusion criteria: the patients confirmed having HIV infection and had not previously received antiretroviral therapy, were asymptomatic, and were aged 18 years or older at the time of recruitment. Unstimulated saliva was collected using the spitting method [19]. Serum samples were also collected. All samples were stored in a refrigerator and frozen at −85 °C until use. The presence of HHV members in the saliva and serum samples were detected using the Clart^®^Entherpex multiplex PCR DNA microarray kit (Genomica, Coslada, Spain). Each specimen was analyzed in duplicate. The results were positive if both reactions yielded a threshold value (Ct) above the limit of detection for the standard. Reactions that produced one positive and one negative result were repeated in duplicate. The samples were positive only when both repeated reactions yielded positive results. Viral genome copy number results were reported as the average of two runs.

## 3. Results

### 3.1. Characteristics of the Study Subjects

A total of 30 subjects were enrolled, consisting of 23 (76.67%) men and 7 (23.33%) women. The majority of subjects (93.33%) were within the age range of 25–49 years. Table 1 shows an overview of the subjects’ characteristics.

### 3.2. The Presence of Human Herpesvirus Members in Sera and Saliva

The present study showed that one or more HHV members were detected in the saliva and sera of all (100%) subjects. In the saliva, EBV (86.67%) was the most frequent co-infection in the HIV-infected individuals, followed by HHV-7 (83.33%), and CMV (63.33%). Meanwhile, in the sera, HHV-7 (76.67%) were the most common in the HIV-infected individuals, followed by CMV (40%) and EBV (30%). VZV and HHV-8 were not detected in either the saliva or sera (Figure 1).

### 3.3. Comparison between the Frequency of Human Herpesvirus Members in Saliva and Sera

The Mann–Whitney test was performed to analyze whether there were differences in the presence of HHV members in the sera and saliva. Table 2 shows that there was no significant difference between several types of HHV members, including HSV-1, HSV-2, VZV, CMV, HHV-7, and HHV-8 (*p* > 0.05).

## 4. Discussion

HHVs can be diagnosed by either detecting the presence of the virus (the whole virus, viral proteins, or genetic material) or antibodies in the blood. Conventional methods and advanced technologies are used. The conventional methods comprise viral culture, serological tests, and molecular techniques using PCR. The PCR method is considered to be the gold standard method due to its sensitivity and specificity in detecting the virus [20]. Multiplex PCR is a type of PCR method that is frequently used for the simultaneous detection and identification of more than one target pathogenic DNA or RNA molecule from one sample using several specific primers in one PCR reaction [21]. Previously, researchers have employed RT-PCR DNA microarrays for the detection of viral pathogens, including HHVs, using different biological samples taken from patients with various diseases [22,23,24]. The multiplex reverse transcription-PCR DNA microarray in our study was used to detect HHVs among HIV-infected individuals who have a high risk of developing chronic and severe HHV-related diseases.

The present study showed that at least one HHV member was detected in both the saliva and sera of all (100%) HIV-infected individuals using an mRT-PCR DNA microarray. Co-infections with two or more types of HHV members were also detected in all the subjects. EBV, HHV-7, and CMV were most frequently detected in the subjects of this study. Similarly, multiple HHVs were also found in all (100%) the serum samples of HIV-infected persons in South Africa, and 100% EBV and CMV were found in 100% of the HIV-infected and ART-naïve adults [2]. Another study conducted in China showed that 89.3% of the HHVs were observed in the peripheral blood samples of individuals with HIV infections under cART; however, the most prevalent HHVs detected among the HIV-positive patients in this study were HSV-2 and HSV-1 [1]. A study on HIV-infected children using paraffin-stimulated saliva demonstrated that 35.4% of the study subjects were positive for HHVs, and CMV was the most commonly detected one [12].

There is evidence to suggest a strong association between both types of HSV and HIV. The interaction between HIV and HSV may play a critical role in HSV-associated disease development, causing more severe persistent symptoms and more frequent recurrences in HIV-/AIDS-infected individuals [25]. Vice versa, HSV-2 infection may enhance the risk of HIV acquisition. Chronic and recurrent HSV-2 infections may cause immune activation and stimulate HIV-1 replication, leading to increases in the levels of HIV RNA (viral load) in patients’ blood and genital secretions and in the risk of HIV progression to AIDS [26,27]. The use of a multiplex reverse transcription-PCR DNA microarray in our study revealed a low detection rate of both HSV-1 and HSV-2 in the saliva (7%, respectively). In the serum samples, HSV-2 was also detected at a low frequency of 7%, while HSV-1 was not found. Consistent with our findings, a previous study utilizing conventional PCR assays reported a similar frequency of HSV-1 detection in saliva (6.2%), but a lower occurrence of HSV-2 (4.2%) among HIV-infected children [12]. However, another study revealed a higher frequency of both types of HSV in HIV-infected people (HSV-2 (65.3%) and HSV-1 (59.5%)) [1]. The various results of HSV detection may be due to the variety of laboratory methods and samples used and the population [28,29]. The accurate laboratory diagnosis of HSV may also depend on other various factors, such as the stage of infection during sample collection, the quality of the specimen, the precision of the chosen technique, and the interpretation of the test results [20]. Future studies focusing on HSV symptomatic patients may provide different outcomes and insights into the characteristics and detection of HSV infections.

The accuracy of laboratory diagnostic tests for detecting HSV infections relies on multiple factors, including the stage of the infection during sample collection, the specimen quality, the specific tests employed, the precision of the method, and the interpretation of the test results by the clinician.

In the present study, VZV was not found in either the sera or the saliva. This result may be due to routine vaccination against VZV, which is usually given to children under 6 years of age or at 12–15 months of age in Indonesia at the same time as the measles–mumps–rubella (MMR) vaccine and the diphtheria–tetanus–pertussis booster, as recommended by the Indonesian Pediatrician Association. Vaccination has proven to be highly effective in reducing the spread of viral infections and enhancing herd immunity [30]. A study reported that a low prevalence of VZV DNA was detected in the saliva (5.1%) [31] and in the peripheral blood of HIV-infected persons (1.7%) [1]. VZV causes varicella (chickenpox) as a primary infection and herpes zoster (shingles) of the orofacial region following the reactivation of latent VZV in the trigeminal nerve. Among HIV-infected individuals, herpes zoster infection may occur during the course of the disease, and this can be considered as an indicator of the disease. VZV also causes meningitis/encephalitis that can lead to significant morbidity at a young age in HIV-infected individuals [32]. A high incidence of herpes zoster (HZ) has been reported in HIV-infected patients with a lower CD4 count, but the risk of HZ has decreased in the cART [33]. It has also been reported that immune reconstitution inflammatory syndrome (IRIS) is associated with an increase in the risk of HZ in the 6 months immediately after the initiation of ART [34]. VZV can also cause acute retinal necrosis (ARN), which is associated with IRIS in HIV patients [35].

Our findings demonstrated that the Epstein–Barr virus (EBV or HHV-4) was the most frequently detected virus in saliva. This result aligns with previous studies that reported a high prevalence of EBV ranging from 90 to 100% among individuals infected with HIV [2,29]. A study reported that the quantity of EBV in the saliva of non-antiretroviral (ARV) HIV- or acquired immunodeficiency syndrome (AIDS)-infected patients was higher (100%) than that of the ARV HIV-/AIDS-infected patients [14]. In the HIV-infected individuals, EBV is associated with a number of oral diseases, including oral hairy leukoplakia (OHL), which was first described by Greenspan et al. in 1984. OHL manifests as a bilateral white patch with a vertical corrugation pattern on the lateral border of the tongue and was described after the emergence of HIV/AIDS [36]. A diagnosis of OHL can be made via EBV DNA detection in the whole saliva, which allows both HIV-infected individuals receiving ART and treatment-naïve patients to receive a more effective treatment. The early screening for EBV DNA or its protein in patients with HIV/AIDS using several diagnostic methods for EBV detection, including serological and molecular diagnostic methods, may reduce the risk of EBV-associated diseases [37,38]. The present study showed that the quality of EBV in the saliva was significantly higher than that in the sera. It has been hypothesized that the high rates of infected B cell activation and the poor cellular immune control of EBV may lead to higher EBV viral loads in the saliva of HIV-infected individuals [39].

Similar to EBV, CMV (HHV-5) is another HHV member virus detected among HIV-infected individuals. The relationship between CMV and HIV has been intensively studied, demonstrating that HIV may synergistically contribute to the spread and infection of both HIV and CMV [4,10,11]. HIV may increase the risk of CMV co-infections and/or reactivation that can cause a wide range of clinical manifestations from asymptomatic to severe symptoms in HIV-infected individuals, including those in the oral cavity, such as oral ulceration, salivary gland hypofunction, and xerostomia [40,41]. Vice versa, the rate of CMV shedding in HIV-infected individuals can increase rapidly, which may contribute to accelerated HIV progression to AIDS. In the cART era, CMV is a significant co-factor in HIV disease that may contribute to HIV acquisition and increase the risk of HIV-related morbidity and mortality due to immune activation and systemic inflammation [42,43,44,45,46,47]. Therefore, screening for CMV by detecting CMV DNA in saliva or blood is highly recommended for HIV/AIDS patients. Several methods have been used to detect CMV in various samples. Our findings showed that CMV was more frequently detected in saliva than it was in the sera, but there was no statistically significant difference. These results were similar to those reported in a previous study [48]. It is believed that the salivary glands are a major site for CMV persistent replication and latency; consequently, the prolonged secretion of virus in saliva is significantly associated with a high risk of CMV transmission [49]. Therefore, exposure to saliva containing infectious CMV is assumed to be the primary route of CMV spread. On the other hand, saliva can be used to diagnose CMV infections as it is easily collected using non-invasive techniques.

Our study showed that there was a significant difference in the quantity of HHV-6 in the saliva and sera in HIV-infected patients, which was detected in the saliva (12%), but not in the sera. It has been identified that the quantity of HHV-6 detected in saliva is due to the salivary glands being a major site for HHV-6 replication and persistence, making saliva a potential transmission route for HHV-6 [50,51]. Another study showed that a low HHV-6 frequency was detected on the surfaces of the tongues of adults (1.4%) using PCR. For HIV-infected children, a study in Brazil reported that HHV-6 was detected in 18% of the oral mucosa [13]. Other studies using plasma samples have reported that the frequency of HHV-6 ranges from 7% to 32.2% [1,52,53]. HHV-6 infection/reactivation in HIV/AIDS individuals causes disseminated infections in many organs, leading to a variety of diseases, including encephalitis, pneumonitis, multiple sclerosis, non-Hodgkin’s and Hodgkin’s lymphomas, and other serious diseases [3,54]. In the oral cavity, HHV-6 has been associated with various oral lesions, such as oral lichen planus, leukoplakia, and oral squamous cell carcinoma [55,56]. In HIV-infected patients, HHV-6 has been detected in 71% of gingival biopsies, suggesting that there is a relationship between the virus and HIV-associated periodontal disease [57].

Similar to HHV-6, the existence of HHV-7 in saliva has also suggested that salivary glands are the major sites of HHV-7 persistent infections [51]. We found that the frequency of HHV-7 was higher than that of HHV-6 in the saliva (83,33%). Our findings were almost similar to a prior study that showed that HIV-infected patients harbored HHV-7 with a high frequency (81%), suggesting that immunosuppression in this population increases the frequency of detection and the viral load of HHV-7 [58]. It has been reported that low CD4 counts were correlated with HHV-7 infections [1]. We also found a high HHV-7 frequency (76.67%) in the sera samples, which is nearly the same as that in a previous study, demonstrating a high prevalence of HHV-7 (86%) observed in HIV-infected individuals [3]. Another study showed HHV-7 was detected in the oral mucosa of 68% of HIV-infected children [13]. Ren et al. (2020) reported a low frequency of HHV-7 (15.6%), but it was higher than that of the healthy controls [1]. HHV-7 has been associated with several diseases both in immunocompromised and immunocompetent persons, including febrile seizures, encephalitis or encephalopathy, and cutaneous diseases such as lichen planus, etc. [59,60]. In the oral cavity, it has been suggested that HHV-7 is associated with active periodontitis [61].

Human herpes virus-8 (HHV-8), also named Kaposi’s sarcoma-associated herpesvirus (KSHV), was also not detected in our study. Previous studies have reported a lower prevalence of HHV-8 infection in Asian countries compared to those in African and other countries, and oral samples have been consistently shown to have the highest rates of HHV-8 viral detection compared to those of other clinical samples, suggesting that saliva may play an important role in HHV-8 transmission [1,16,62,63]. HHV-8 is considered to be the etiological pathogen of Kaposi’s sarcoma (KS) in HIV-infected individuals with low CD4+ T cell counts and high viral loads [3]. However, the majority of individuals with HHV-8 infection do not show any symptoms. The lack of a gold standard for determining the HHV-8 serological status further complicates the ability to accurately determine the true infection status as well. Therefore, a negative test result may not definitively exclude the presence of HHV-8. The results should be combined with the patient’s clinical profile for comprehensive evaluation.

HHV members are highly contagious pathogens that may accelerate the progression of HIV infection to AIDS. The detection of HHVs is important for clinical diagnoses and treatments because HHVs infections may not be accurately diagnosed during clinical examination; therefore, laboratory diagnoses should be conducted. This study highlights the utility of multiplex PCR as a valuable diagnostic tool for HHVs infections, enabling the simultaneous detection of all HHV members in various clinical samples. However, this study has some limitations. We did not specifically validate the kit for saliva, but a previous study conducted by Druce et al. successfully validated a multiplex-PCR microarray-based method using various clinical specimens, including saliva [64]. Another limitation of our study is that we did not investigate the presence of HHVs in the saliva and sera of HIV-positive patients who were undergoing antiretroviral therapy (cART) or consider the immune status of subjects who have clinical symptoms. Accordingly, future studies must assess the correlation between the presence of HHVs and their clinical manifestation before and after cART initiation to achieve a better understanding of how the reactivation of members of HHVs may cause HIV disease progression. Future studies with a larger sample size for more precise results are also needed.

## 5. Conclusions

The findings of this study demonstrate that, according to multiplex PCR, EBV and CMV are significantly more prevalent in saliva than they are in sera. Other HHV members, HSV-1, EBV, CMV, HHV-6, and HHV-7, were also more prevalent in saliva than in sera among asymptomatic individuals with HIV, but the results were not significantly different. The implementation of multiplex PCR as a diagnostic tool offers significant value, enabling the simultaneous detection of multiple viral pathogens within a single clinical sample. This approach can be employed as an effective screening or first-line test for detecting HHV infections.

## Figures and Tables

**Figure 1 pathogens-12-00993-f001:**
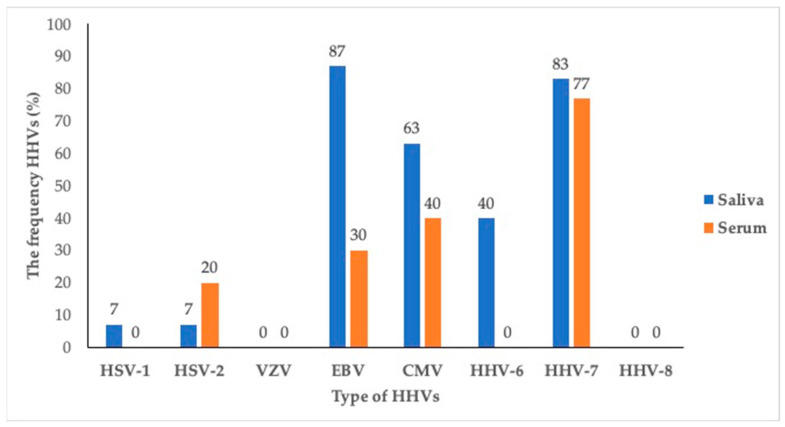
The frequency of human herpesvirus members in sera and saliva of asymptomatic HIV-infected individuals.

**Table 1 pathogens-12-00993-t001:** Characteristics of study subjects.

Variables	N = 30		*p*-Value
	Frequency	Percentage	
Gender			
Male	23	76.67%	0.006 *
Female	7	23.33%	
Age (years)			
20–24 year	1	3.33%	0.000 *
25–49 year	28	93.33%	
≥50 year	1	3.33%	
Mean + standard deviation	34.8 + 8.62		

Analysis using Mann–Whitney test; * significant *p* < 0.05.

**Table 2 pathogens-12-00993-t002:** Comparison between the frequency of HHV members detected in saliva and sera of asymptomatic HIV-infected individuals.

No	Types of HHVs	Saliva	Serum	*p*-Value
		N (%)	*n* (%)	
1	HSV-1	2 (7%)	0 (0%)	0.154
2	HSV-2	2 (7%)	6 (20%)	0.132
3	VZV	0 (0%)	0 (0%)	1.000
4	EBV	26 (87%)	9 (30%)	0.000 *
5	CMV	19 (63%)	12 (40%)	0.073
6	HHV-6	12 (40%)	0 (0%)	0.000 *
7	HHV-7	25 (83%)	23 (77%)	0.522
8	HHV-8	0 (0%)	0 (0%)	1.000

Analysis using Mann–Whitney test; * significant *p* < 0.05.

## Data Availability

The data that support the findings of this study are available on request from the corresponding author.

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
