# Peer review of "Detection of Human Herpesviruses in Sera and Saliva of Asymptomatic HIV-Infected Individuals Using Multiplex RT-PCR DNA Microarray"

_pathogens, 2023, doi:10.3390/pathogens12080993_

Round 1

Reviewer 1 Report

The article explained prevalence of HHVs in saliva and serum of asymptomatic HIV-infected individuals. The authors fully described The critical role of HHVs as co-pathogens in disease progression of HIV infection. The authors put emphasis on the  multiplex PCR as a valuable diagnostic tool for the simultaneous detection of viral pathogens. Discussion section is exhaustive.

I would like to suggest an improvement for review:

-line 25-28:

''There was no significant difference between HHVs in salivary (75%) and serum (50%) (p>0,05)''

Why did you write "Mixed infections of HHVs were found in 90% of saliva and 70% of serum samples "? The percentage of positive results  in saliva and serum is different. Please explain.

- line 32: According to Table 3:  EBV and HHV-6  were significantly more prevalent in saliva than in serum, Please check for typos.

- line 31- 34

According to article: EBV and CMV were significantly more prevalent in saliva than in serum of asymptomatic HIV-infected  individuals  with p≤0.005, so sentence- " HSV-1, EBV, CMV, HHV-6, and HHV-7 were more abundant in saliva than in the serum" is not true.  According to Table 3: The comparison between saliva and serum for HSV-1, CMV, HHV-7 is statistically insignificant. In my opinion, second sentence should be rejected or corrected.

-line 98: figure 1 should be added

- line 104;  The signature of Y- axis in Figure 2 should be corrected. In my opinion, should be ''The frequency of HHVs (%) " .

- line 110: should be HSV-1 and HSV-2. Please check for typos.

- line 145: 3 ?, please review your sentence

-line 164-165: virus (VZV) was found either in serum or saliva?, please review your sentence. According to study, "VZV and HHV-8 were not detected in either the saliva or serum"?

- line 172: please explain abbreviation

-line 178-180 the sentence is not clearly, please review your sentence

-line 226, should be HHV-6

Author Response

Dear Reviewer,
Thank you for giving us the opportunity to submit a revised draft of the manuscript “Detection of Human Herpesviruses in Serum and Saliva among Asymptomatic HIV-Infected Individuals by Multiplex Reverse Transcription-PCR DNA Microarray” for publication in the Journal of Pathogens. We appreciate the time and effort that you and the reviewers dedicated to providing feedback on our manuscript and are grateful for the insightful comments on and valuable improvements to our paper. We have made the changes directly to the manuscript and in the form of a response to Reviewer 1 Comments (attached).

Sincerely,

Irna Sufiawati

Reviewer 2 Report

The experiment reported in this manuscript would seem to advance our knowledge of HIV and HHV co-infection in three ways: use of a multiplex assay to detect HHV in saliva, comparison of saliva to serum, and assessment of a cohort composed of HIV-infected but asymptomatic and untreated individuals.  However, only the first two are articulated in the abstract and at the end of the introduction.  The result demonstrates that some HHV are more prevalent than others in saliva of this patient cohort and two others were not detected.

General comments:

1   1) More could be done with the Results section.  Paired saliva and serum samples were assayed.  Do individuals who have a given HHV detected in serum also have that same virus detected in their bloodstream?  This information would go a long way toward addressing why the authors chose to compare saliva and serum.

2   2) Has the multiplex assay been used to screen healthy, HIV-negative persons?  If such information exists (here or in the literature) it should be described as it would go a long way toward addressing whether the HIV-positive, asymptomatic and untreated patient population is already more susceptible to HHV reactivations.

3   3) The extensive discussion belabors the results of the herpesviruses one by one compared with existing literature rather than focusing on what has been gained by the unique aspects of the study.  The discussion should be trimmed significantly and refocused.  Some of the discussion, in much abbreviated form, may be able to shift to the introduction.

         4) It seems the ultimate goal of this work is an assay that detects HHV in saliva so treatment can be initiated to prevent development of oral diseases (lines 63-65).  However, it should be acknowledged that currently there is no standard antiviral treatment for HHV-6A, HHV-6B or EBV, three of the four most prevalent viruses in their study.

Specific points:

1   1) The topic sentence in lines 49-52 seems misleading as the hole in the literature since this patient cohort was not being treated at the time of sample collection.

2   2) The Y axis in Fig. 2 is incorrect English.  It should read “Frequency of HHV (%)” or “Percent positive for HHV.”

3   3) Line 165: The results demonstrate that VZV was found in neither serum nor saliva (rather than “either in serum or saliva”).

4   4) Discussion points: What was the VZV vaccination status of this young cohort (97% aged 20-49 yr)?  Could vaccination explain the lack of detection of VZV?  The Clart Entherphex kit (note misspelling on line 82) does not appear to have been validated for saliva.  This should be acknowledged as a limitation and possible explanation for no detection of VZV and HHV-8, and low detection of HSVs.

5   5) Things are overstated in several places (e.g. lines 261-263). 

In certain paragraphs the English is excellent (e.g. lines 66-88 and most of the Discussion).  Most of the Introduction and Results should be re-worked to improve comprehensibility.

Author Response

Thank you for giving us the opportunity to revise our manuscript entitled “Detection of Human Herpesviruses in Serum and Saliva among Asymptomatic HIV-Infected Individuals by Multiplex Reverse Transcription-PCR DNA Microarray,” for publication in the esteemed journal, Pathogens. We highly appreciate the Reviewers for the precious time and effort dedicated to reviewing our paper and providing valuable comments. The authors have carefully considered the comments and tried our best to address every one of them. 

In the form of a Response Letter of Reviewers Comments, we provide a point-by-point response regarding the changes made to the manuscript, red highlighted text indicates where changes have been made. 

We hope the revised version is now suitable for publication.

Sincerely,

Irna Sufiawati
